# Study on the Synthesis and Thermal Stability of Silicone Resin Containing Trifluorovinyl Ether Groups

**DOI:** 10.3390/polym12102284

**Published:** 2020-10-05

**Authors:** Rui Huang, Jinshui Yao, Qiuhong Mu, Dan Peng, Hui Zhao, Zhizhou Yang

**Affiliations:** 1School of Materials Science and Engineering, Qilu University of Technology (Shandong Academy of Sciences), Jinan 250353, China; ruihuang258@163.com (R.H.); yaojsh@qlu.edu.cn (J.Y.); kinglmq@163.com (Q.M.); lonarpeng@aliyun.com (D.P.); huizhaochem@163.com (H.Z.); 2Shandong Provincial Key Laboratory of Special Silicone-Containing Materials, Advanced Materials Institute, Qilu University of Technology (Shandong Academy of Sciences), Jinan 250014, China; 3School of Chemical Engineering, Sichuan University, Chengdu 610065, China

**Keywords:** silicone resin, trifluorovinyl ether group, thermal stability, degradation

## Abstract

Silicone resin is a high-temperature resistant material with excellent performance. The improvement of its thermal stability has always been the pursuit of researchers. In this paper, a sequence of silicone resins containing trifluorovinyl ether groups were prepared by the co-hydrolysis-polycondensation of methyl alkoxysilane monomers and {4-[trifluorovinyl(oxygen)]phenyl}methyldiethoxysilane. The structures of the silicone resins were characterized by FT-IR and ^1^H NMR. The curing process of them was studied by DSC and FT-IR spectra, and results showed that the curing of the resins included the condensation of the Si-OH groups and the [2 + 2] cyclodimerization reaction of the TFVE groups, which converted to perfluorocyclobutane structure after curing. The thermal stability and thermal degradation behavior of them was studied by TGA and FT-IR spectra. Compared with the pure methyl silicone resin, silicone resins containing TFVE groups showed better thermal stability under both N_2_ and air atmosphere. Their hydrophobic properties were characterized by contact angle test. Results showed that PFCB structure also improved the hydrophobicity of the silicone resin.

## 1. Introduction

Silicone resin is a highly crosslinked polysiloxane with a three-dimensional network, which is usually prepared by hydrolytic polycondensation of polyfunctional organosilanes [1,2,3]. They have many excellent properties, such as thermal stability, oxidation resistance, etc [4,5]. Therefore, silicone resins have many applications, such as thermal protection materials, engine sealant, high temperature-resistant adhesives, and light-emitting diode chips, etc [6,7,8]. Commonly, silicone resin can be used for a long time below 250 °C, mainly because its Si–O bond (about 460.5 kJ/mol) backbone has higher energy than the C–O bond (358.0 kJ/mol) and the C–C bond (304.0 kJ/mol) [9].

The development of high-performance polymers for aerospace has always been an important area of polymer science. The search for materials with high thermal stability is a goal of this research. Silicone resin has always been a good choice for high thermal stability materials. It has always been the goal of researchers to further improve the high temperature resistance of silicone resins. In order to meet higher requirements, researchers have proposed many methods to enhance the thermal stability of them. SiO_2_ nanoparticles [10] and Al_2_O_3_ [11] could enhance the stability of silicone resins as fillers. The replacement of the pendant methyl groups by phenyl groups was also a conducive method to improve their thermal stability [3]. Silphenylene units [12] and boron atom [13] were also introduced into the skeleton of silicone resins for their thermal stability improvements. As shown in Scheme 1, the back-biting degradation caused by Si-OH groups at the end of silicone resins at high temperature [14,15] was the main reason for the degradation of them. Recent studies found that POSS [16,17,18] could react with terminal Si-OH, thus eliminating back-biting depolymerization. However, these improvements were limited to the method of changing the structure of silicone resins and using additives. Therefore, the development of new methods to improve the thermal stability of silicone resins is of great significance to their application.

In addition to using additives and changing the main chain structure, the introduction of cross-linkable functional groups on the side chain is also an effective method to improve the thermal stability of silicone resins. For this method, the thermal stability of the cross-linked structure formed by the side chain functional groups has an important influence on the improvement of the thermal stability of the silicone resins. Therefore, the choice of side chain functional groups is very important. Previous research has shown that trifluorovinyl ether (TFVE) groups could not only be easily attached to polymers, but also could be converted into perfluorocyclobutane (PFCB) structures with excellent thermal stability through {2 + 2} ring polymerization [19]. Up to now, TFVE groups have been widely used in various materials to improve their thermal stabilities. Zhou et al. [20] introduced TFVE group into phenolic resin and improved its thermal stability through post-crosslinking. In addition, TFVE groups have also been introduced into other polymers, such as poly(arylene ether) [21], poly(fluorene-co-sulfone)ether [22], phenylphosphine oxide materials [23] and so on. All the obtained materials exhibited excellent thermal stability. Recently, Yuan et al. [24] also introduced it into linear polysiloxane to improve their thermal stability. For branched silicone resins, the introduction of TEVE groups should also be a very good choice to improve their thermal stability.

The purpose of this work was to improve the thermal stability of silicone resins by the introduction of TFVE groups. In this work, a sequence of silicone resins containing different contents of TFVE groups was synthesized. The curing process of the new silicone resins was investigated by DSC and FT-IR. The effect of the amount of TFVE groups on the thermal stability and the degradation process of the silicone resin was investigated. This study could provide a new strategy for the research of high temperature-resistant silicone resins, and the research on the curing mechanism and degradation process could provide theoretical support for the application of this kind of silicone resin.

## 2. Materials and Methods

### 2.1. Materials

4-bromophenol was purchased from Shanghai Jiuding Chemical Technology Co., Ltd. (Shanghai, China), and 1,2-dibromotetrafluoroethane was purchased from Sichuan Shangfu Technology Co., Ltd. (Sichuan, China). Methyl triethoxysilane (MTES) and dimethyl diethoxysilane (DMDES) were purchased from Shanghai Mclean Biochemical Technology Co., Ltd. (Shanghai, China). Other materials were purchased from Shanghai Aladdin Biochemical Technology Co., Ltd. (Shanghai, China). Unless otherwise specified, all chemical reagents are used without further purification.

### 2.2. General Characterization

^1^H NMR, ^19^F NMR and ^13^C NMR spectra were measured on AVANCE II 400 instrument (Bruker, Billerica, MA, USA). FT-IR spectra were recorded on a Nicolet spectrometer (Nicolet, Thermo Fisher Scientific, Waltham, MA, USA) with KBr pellets. Differential scanning calorimetry (DSC) analysis was performed using a differential scanning calorimeter (DSC Q10, TA Instruments, New Castle, DE, USA) at a heating rate of 10 °C/min and a N_2_ flow rate of 50 mL/min. Thermogravimetric analysis (TGA) was measured on a thermogravimetric analyzer (Mettler Toledo, Columbus, OH, USA) at a heating rate of 10 °C/min. The water contact angle was measured by DSA25 contact angle tester (KRUSS, Hamburg, Germany).

### 2.3. Synthesis of the Intermediates and the Silane Monomer

The synthesis of the intermediates and the silane monomer is shown in Scheme 2. The preparation of these compounds was based on the previously reported method [19,24,25].

#### 2.3.1. Synthesis of 4-[2-Bromotetrafluoroethoxy]bromobenzene (M1)

To a 1 L, three-neck flask equipped with a condenser and a magneton was added 4-bromophenol (86.50 g, 0.50 mol), m-xylene (85 mL), dimethylsulfoxide (350 mL), and cesium carbonate (179.20 g, 0.55 mol). The mixture was heated to 120 °C. After the cesium carbonate dissolved, the azeotrope of m-xylene/water in the reaction was removed. The reaction was kept at 120 °C for 36 h and then cooled to room temperature. After that, 1, 2-dibromotetrafluoroethane (142.90 g, 0.55 mol) was added dropwise and the mixture was stirred at room temperature for 4 h, then the temperature was raised to 50 °C and kept for 16 h. One liter of distilled water was added to the mixture and then extracted with dichloromethane. The organic phase was washed with 5% sodium carbonate solution and distilled water and then dried with anhydrous sodium sulfate and filtered. The crude product was obtained by distillation, which was purified by chromatography using n-hexane as eluent to afford 89.72 g (51.0%) of M1 as a colorless liquid. ^1^H NMR (CDCl3), δ (ppm): 6.95 (2 H, aromatic ring) and 7.43 (2 H, aromatic ring); FT-IR: 1484, 1205, 1164, 1130, and 941 cm^−1^.

#### 2.3.2. Synthesis of 4-[Trifluorovinyl(oxygen)]bromobenzene (M2)

Activated zinc powder (20.47 g, 0.31 mol) and acetonitrile (160 mL) were added into a 250 mL three-neck flask and stirred at room temperature, then M1 (87.95 g, 0.25 mol) was dropped into the mixture. The mixture was heated to reflux and maintained for 24 h. After cooling to room temperature, acetonitrile was removed by distillation, then n-hexane was added to extract the product. The mixture was filtered and concentrated to afford a yellow liquid. The above liquid was distilled under reduced pressure to obtain 40.28 g (63.7%) of a colorless liquid. ^1^H NMR (CDCl3) δ (ppm): 6.95 (2 H, aromatic ring) and 7.43 ppm (2 H, aromatic ring); FT-IR: 1833 (CF=CF_2_), 1484, and 825 cm^−1^.

#### 2.3.3. Synthesis of {4-[Trifluorovinyl(oxygen)]phenyl}methyldiethoxysilane (M3)

Magnesium turnings (4.40 g, 0.18 mol), MTES (37.44 g, 0.21 mol), iodine (0.001g), and THF (220 mL) were added into a round bottom flask, which was equipped with a condenser tube and a drip funnel. M2 (40.00 g, 0.16 mol) in dry THF (20 mL) was added dropwise into the round bottom flask at 65 °C. The mixture was stirred at this temperature for 24 h. Then THF was removed by evaporation, and 100 mL of n-hexane was added to the mixture to extract the product. The solid was removed by filtration, and the filtrate was concentrated by distillation. The crude product was distilled under reduced pressure to obtain 21.50 g (44.4%) of a colorless liquid. ^1^H NMR (CDCl3), δ (ppm): 0.36 (3H, Si-CH3), 1.25 (6H, O-CH2-CH3), 3.76–3.87 (4H, O-CH2-CH3), 7.13 (2H, aromatic ring), 7.65 (2H, aromatic ring); ^19^F NMR (CDCl3), δ (ppm): −134.1 to −134.4, −126.2 to −126.9, −119.8 to −120.8; ^13^C NMR (CDCl3), δ (ppm): −4.17, 18.26, 58.56, 116.08, 131.65, 135.27, 143.92, 146.71, 147.40, 156.51. FT-IR: 2973, 1833 (CF=CF_2_), 1592, and 1280 cm^−1^ (Si-CH3).

### 2.4. Synthesis of the Silicone Resins

The methyl silicone resin (M-SR) and the silicone resins containing TFVE groups (F-SRs) were synthesized by the co-hydrolysis-polycondensation method [24]. The synthetic route was shown in Scheme 3. According to the content of TFVE groups, the samples were named as M-SR, F-SR-1, F-SR-2, F-SR-3, and F-SR-4. Detailed synthesis data are listed in Table 1. As shown in Table 1, the ratio of organic groups to silicon atoms (R/Si) of all silicone resins was 1.5. R_F_/R was the ratio of TFVE groups to all the organic groups. Take the synthesis of F-SR-2 as an example. A mixture of MTES (2.67 g, 15 mmol), DMDES (1.11 g, 7.5 mmol), M3 (2.30 g, 7.5 mmol), and glacial acetic acid (18.00 g, 0.30 mol) was heated to reflux and kept at this temperature for 24 h. After cooling to room temperature, the solvent and small molecular substances were removed by evaporation. Then 50 mL of ethyl acetate was added. The mixture was washed with distilled water to remove the residual acid and dried with anhydrous calcium chloride. Finally, ethyl acetate was evaporated by vacuum to give a yellow, viscous liquid (3.23 g, yield: 82.7%).

### 2.5. Curing of the Silicone Resins

The curing of silicone resins are shown in Figure 1. The silicone resin was put into a crucible, then the crucible was placed in an oven for gradient heating and curing. The temperature was raised stepwise and maintained at 90 °C for 0.5 h, 120 °C for 1 h, 150 °C for 1 h, 180 °C for 1 h, 210 °C for 1 h, 240 °C for 1 h, and 270 °C for 2 h respectively.

## 3. Results and Discussion

### 3.1. Characterization of the F-SRs

The FT-IR spectra of the F-SRs with different contents of TFVE groups are shown in Figure 2. All silicone resins showed a broad peak near 1000 cm^−1^ belonging to Si–O–Si bond, which indicates the formation of Si–O–Si main chain in the silicone resins. The characteristic peak near 1260 cm^−1^ indicated the presence of Si–CH_3_, and the peak at 2969 cm^−1^ belonged to the C–H bond of Si–CH_3_ groups. The formation of Si–OH groups in the silicone resins could be confirmed by the characteristic peak at 3400 cm^−1^. Compared with M-SR, F-SRs showed a characteristic absorption peak at 1833 cm^−1^ belonging to the vibration of CF=CF_2_ of TFVE groups. Moreover, the intensity of this absorption peak gradually increased with its content. Correspondingly, the absorption peak at 3010 cm^−1^, which belonged to the C-H bond of the benzene ring in TFVE groups, also gradually increased.

To further confirm the structure of F-SRs, ^1^H NMR measurement was also used for their structure confirmation. As shown in Figure 3, the characteristic signals of 7.6 and 7.2 ppm belonged to the benzene rings in TFVE group, and the signal at 0–0.5 ppm was due to the hydrogen of Si–CH_3_. Moreover, the characteristic signal of Si–O–CH_2_CH_3_ was not observed at 1.2 and 3.8 ppm, indicating that the hydrolysis reaction was complete. Additionally, the actual values of R_F_/R calculated by nuclear magnetic data were close to the theoretical values (Table 1). This further indicated that F-SRs were successfully synthesized as expected.

### 3.2. Curing Behavior of the F-SRs

DSC was used to study the curing behavior of this kind of silicone resin, and the results are shown in Figure 4. Compared with the methyl silicone resin (M-SR), other F-SRs showed obvious exothermic peaks in the temperature range of 160 °C to 300 °C. As described in previous reports, the peak was attributed to the cyclization reaction of TFVE groups [25,26,27]. Additionally, the exothermic peaks of silicone resins with different TFVE group contents were slightly different. With the increase of the TFVE group contents, the temperature corresponding to the maximum peak position gradually decreased. In the curing process, the cyclization reaction required the TFVE groups to be close to each other. As a result, the reaction became easier with the increase of the content of TFVE groups. In order to confirm the degree of the cyclization reaction, the cured silicone resins were retested by DSC. As shown in Figure 4b, the exothermic peaks on DSC curves disappeared [25,28], which indicated that the cyclization reaction of the TFVE groups was complete.

To further investigate the structural changes of F-SRs after curing, the FT-IR spectra of the cured resins were tested and shown in Figure 5. Compared with the uncured silicone resin, the characteristic peak of CF=CF_2_ at 1833 cm^−1^ disappeared and the characteristic peak of the PFCB ring structure appeared at 961 cm^−1^ in the cured one, indicating the occurrence of cyclization reaction and the formation of the PFCB structure. Moreover, the characteristic peak of PFCB ring became obvious with the increase of the TFVE content, which was consistent with the previous analysis. In addition, the characteristic peak of Si–OH around 3400 cm^−1^ disappeared in the cured F-SRs, which demonstrated that F-SRs also underwent dehydration condensation reaction of Si–OH during the curing process like common silicone resins.

### 3.3. Thermal Properties and Degradation of F-SRs

The thermal stability of silicone resins was studied by TGA. The TGA and DTG curves of them under N_2_ atmosphere are shown in Figure 6. Table 2 showed the characteristic data of all these curves. It could be seen that with the increase of TFVE group content, the temperature for 5% mass loss (T_5%_) increased from 374 °C to 461 °C and the temperature for 10% mass loss (T_10%_) increased from 415 °C to 500 °C, which was comparable to other silicone resins that have been reported [9,12,17]. From the DTG curves (Figure 6b), it could be found that the degradation of silicone resins was mainly divided into two stages, T_5%_ was in the first stage range. Additionally, the temperatures for the maximum degradation rate in the first stage (T_max1_) of the F-SRs (F-SR-1, F-SR-2 and F-SR-3) were higher than that of M-SR, and this stage did not exist for F-SR-4. According to the previous reports, the degradation in this stage was mainly due to the back-biting reaction caused by the residual Si–OH groups in the cured silicone resins [29,30,31]. The increase of T_5%_ and T_max1_ indicated that the formation of the PFCB structure inhibited the occurrence of back-biting reaction and improved the thermal stability of F-SRs. As the back-biting reaction was suppressed, the degradation of F-SRs at this stage was not obvious. This could be confirmed by the FT-IR spectra of the silicone resin after heat treatment at this stage (Figure 7). The FT-IR spectrum of F-SR-4 treated at 400 °C was almost unchanged compared with that of the untreated one, indicating that the structure of it was stable at this temperature. The temperature of the maximum mass loss rate in the second stage (T_max2_) was in the range of 537–547 °C, which was much higher than that of M-SR (490 °C). The degradation at this stage was mainly caused by the main chain breakage. As the PFCB structure had high thermal stability and the increase of the crosslinking degree could inhibit the thermal degradation of the main chain, the thermal stability of F-SRs at this stage was greatly improved. As shown in Figure 7, the characteristic peak of Si–CH_3_ groups of F-SR-4 was still obvious after being treated at 540 °C, which indicated that the existence of PFCB structure greatly inhibited the degradation of the silicone resin. The improvement of the thermal stability at this stage also led to an increase of the char yield. Correspondingly, the char yield of silicone resins increased from 10.8% to more than 50%. After the second stage, the structure of F-SRs was severely damaged at higher temperature. As shown in Figure 7, the characteristic peaks of organic groups in the FT-IR spectra of F-SR-4 treated at 800 °C completely disappeared.

TGA and DTG curves of all the silicone resins in air atmosphere are shown in Figure 8. The detailed data are shown in Table 3. With the increase of TFVE groups content, T_5%_ increased from 362 °C to 392 °C and T_10%_ increased from 389 °C to 447 °C. According to the DTG curves, the degradation of them under air atmosphere included three stages. At the first stage, the temperature was not very high, and the structure of the silicone resin was relatively complete, so the dissolution and diffusion of oxygen in the system were not easy. It was believed that the degradation caused by the back-biting reaction was still the main reason for the degradation of this stage rather than the oxidation reaction. The degradation of the second stage was mainly caused by the oxidation reaction, which was mainly affected by the dissolution and diffusion of oxygen in the material [32]. The cross-linking degree and the intermolecular force of F-SRs had been improved due to the formation of PFCB structure, so the dissolution and diffusion of oxygen in F-SRs became more difficult compared to M-SR. Therefore, the temperature for maximum mass loss rate of the F-SRs (492–502 °C) was much higher than that of M-SR (392 °C). As shown in Figure 9, F-SR-4 still contained a small amount of organic groups after being treated at 500 °C, which was further proof that the oxidative degradation of F-SRs was inhibited. Thereafter, the remaining organic groups were further oxidized at the higher temperature. Finally, the silicone resin was completely oxidized to silica. As a result, the char yield of them decreased from 73.2% to 32.6% with the increase of the TFVE amount. Compared with other reported silicone resins [3,15], F-SRs showed better thermal stability in air atmosphere.

### 3.4. Hydrophobicity of the F-SRs

Hydrophobicity can give materials some new additional functions, on the basis of which many new surface technologies can be developed. Because F-SRs contain a large amount of elemental fluorine, this makes them a basis for hydrophobic materials. The hydrophobicity of the cured F-SRs was measured by water contact angle test, and the results are shown in Figure 10. The water contact angle of the silicone resins increased from 113.8° to 128.3° with the increase of TFVE content. This indicated that, to a certain extent, the hydrophobicity of F-SR could be controlled by the content of PFCB structure.

## 4. Conclusions

In summary, a series of silicone resins containing TFVE groups have been successfully prepared by a co-hydrolysis-polycondensation method. During the curing process, TFVE groups in F-SRs underwent [2 + 2] cyclodimerization reaction and formed PFCB structure. The effect of PFCB structure on the thermal stability of the silicone resin was studied. The contact angle test showed that the water contact angle of F-SRs was up to 128.3°, which was a potential hydrophobic material that could be used in the fields of waterproof, anti-fogging, self-cleaning, etc.

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
