# Peer review of "Study on the Synthesis and Thermal Stability of Silicone Resin Containing Trifluorovinyl Ether Groups"

_polymers, 2020, doi:10.3390/polym12102284_

Round 1
Reviewer 1 Report
The paper of Huang et al. describes the synthesis and characterization of modified silicone resins with focus on their thermal stability and is well in scope of Polymers. This work is scientifically sound, but the originality, significance and overall merit are not clearly described in my opinion. Therefore, my overall recommendation is to reconsider after major revisions.
Broad comments
- Abstract: in my opinion the abstract should be rewritten by carefully following the directions of the journal: “Abstract: The abstract should be a total of about 200 words maximum. The abstract should be a single paragraph and should follow the style of structured abstracts, but without headings: 1) Background: Place the question addressed in a broad context and highlight the purpose of the study; 2) Methods: Describe briefly the main methods or treatments applied. Include any relevant preregistration numbers, and species and strains of any animals used. 3) Results: Summarize the article's main findings; and 4) Conclusion: Indicate the main conclusions or interpretations. The abstract should be an objective representation of the article: it must not contain results which are not presented and substantiated in the main text and should not exaggerate the main conclusions.”. In particular, point 1 (question and purpose of the study) is not described.
- Introduction: What are the applications of silicone resins? Why is improved thermal stability of interest? There is only a very general sentence that does not really explain the rationale and the novelty of this work. What is it going to add on the existing knowledge in the field and how is it expected to impact its progress? Are there more types of resins where this approach could be applied? This could potentially increase the significance of this work.
- A very similar work has already been published in Macromolecules (doi: 10.1021/ma501263c) and the authors need to elaborate on how their work is different than this one. How do the results reported herein provide an advance in the already published data?
Specific comments
- Abbreviations should be explained the first time they are used (e.g. TFVE in the abstract) and should be the same throughout the manuscript, for example in the abstract both “FT-IR” and “FTIR” are used. In general though, the use of abbreviations in any abstract should be avoided if not absolutely necessary.
- Correct typo errors, e.g. in N2 the 2 should be subscript, the degree symbol has to be inserted through the symbols in MS word
- Correct formatting errors, e.g. the introduction section is aligned left.
- Grammar needs some editing. For example, its thermal stability not thermal stabilities, even when the objective is in plural.
- Introduction line 42: “these improvements are relatively limited”. Please elaborate on what limited means and why the already used methods for improving thermal stability of silicones arent’ adequate.
- Introduction: “Nowadays, TFVE groups have been widely used in various resin materials to improve their thermal stabilities”. There are only 2 references to support this statement. If only those 2 exist, TFVE aren’t widely used. If there are more, references need to be added and their main findings should be briefly described. What are the advantages of using this modification that led to the decision to perform this study?
- What is the difference of the paper of Yuan et al. (Postpolymerization of Functional Organosiloxanes: An Efficient Strategy for Preparation of Low-k Material with Enhanced Thermostability and Mechanical Properties) with this paper? What improvements are the authors suggesting to their method of attaching TFVE on silicones with post-polymerization? Because of that existing paper, the novelty of the current work is not clear.
- Introduction line 53: “Yuan et al. [18] also 54 introduced it into polysiloxanes to improve their thermal stability. For silicone resins, the introduction of TEVE groups should also be a very good choice to improve their thermal stabilities”. Aren’t polysiloxanes also silicone resins? The way these two sentences are worded implies silicone resins are a different class of materials.
- Are the reactions described in the experimental part adopted from literature? If so, references need to be added.
- Figure 1 is very blurry, most likely because of the conversion to pdf but keep that in mind during revision in case its quality has to be improved.
- Scheme 2. The abbreviation F-SRs is not explained anywhere in the text.
- Table 1. Please explain what the RF/R ratio is. The first column is not a sample number, but abbreviation. Please explain them when they are first mentioned in the manuscript. For example, the explanation of M-SR first appears in section 3.1.
- What was the purpose of using different MTES and DMDES ratios? What changes are the different ratios expected to impart on the final resins?
- Figure 2. Please indicate the mentioned FTIR bands (especially the TFVE) with arrows or any other way the authors prefer. Which bonds’ vibrations cause the peak at 1833 cm-1?
- Section 3.2: Is the peak temperature shift enough of an indication about the “easier” cyclization reaction? What about the curing enthalpy (heat of reaction)? Did it increase or decrease, and what is its physical meaning?
- Section 3.2: Why is it stated that curing was relatively and not fully complete? Is there some enthalpy still detected? If so, can the authors calculate the degree of curing from the DSC data with the formula % cure = 1 - (∆H Residual Cure / ∆H Full Cure) * 100 to support the conclusion that curing was almost fully completed?
- Section 3.2: Was there a Tg detected?
- Section 3.2: which bonds’ vibration causes the FTIR band at 961 cm-1?
- Section 3.3: In my opinion, the temperature where degradation starts (T onset determined using the intersection of tangents to the TGA curve of the pre-degradation region and region of steep decline) could also be reported in table 2. Same for the oxidation onset.
- The intermolecular reaction mentioned is called backbiting and doesn’t need quotation marks. Could the authors add a scheme presenting this reaction?
- Line 223: Personal expressions like "we, our" should be avoided
- Line 227: Experiments, results and conclusions must be reported in past tense (neither present nor perfect tense)
- Fig 10: The values of contact angle are not visible.
- Where do these results stands in comparison with silicones with other functional groups used with the intention to improve their thermal stability? Some discussion must be added to compare these results with other published studies.
Finally, I would like to thank the authors in advance for taking the time to adress my comments.
Reviewer 2 Report
In presented work authors presents how it is possible to improve improve the thermal stabilities of silicone resins by the introduction of trifluorovinyl ether groups (TFVE). The materials and methods section is written very well and provide very detailed information. Results are scientifically sound and well supported by experimental data. I recommend this paper to accept to publish after minor revision.
- Page 1, line7, In abstract and in Introduction (page2, line 49). The used abbreviation TFVE should be explained.
- In Table 1. (Details of the silicone resins synthesis) are presented values of RF/R/Found. In the text should be explained in more datil as the found values of RF/R were calculated.
Yours sincerely
The Reviewer
